# Sexual Violence, Disclosure Pattern, and Abortion and Post-Abortion Care Services in Displaced People’s Camps in Africa: A Scoping Review

**DOI:** 10.3390/ijerph21081001

**Published:** 2024-07-30

**Authors:** Paul O. Adekola, Sunday A. Adedini

**Affiliations:** 1Department of Political Science and International Relations, School of Social Sciences, University of Geneva, 1211 Geneva, Switzerland; 2Department of Population, Family and Reproductive Health, School of Public Health, University of Medical Sciences (UNIMED), Ondo City PMB 536, Nigeria; 3Department of Demography and Social Statistics, Federal University, Oye-Ekiti PMB 373, Nigeria; sunday.adedini@gmail.com; 4Demography and Population Studies Programme, School of Public Health and Social Sciences, University of the Witwatersrand, Private Bag 3, Wits, Johannesburg 2050, South Africa

**Keywords:** sexual violence, abortion, post-abortion care, Africa, internally displaced persons

## Abstract

Violent social and political conflicts have caused several challenges to internally displaced persons (IDPs), especially girls and young women, among which is sexual violence (SV). Despite extensive records on SV in humanitarian contexts, studies to assess the level, examine the disclosure pattern (DP) and evaluate the availability of abortion care in these settings have received inadequate attention. This scoping review sought to synthesise the current African-based research on SV, DP, and abortion and post-abortion care (APAC) in humanitarian contexts. We conducted a systematic search of five databases: MEDLINE, PubMed, Scopus, Embase and Google Scholar, where the articles retrieved met the criteria for inclusion. The review adhered to PRISMA guidelines and the Critical Appraisal Skills Programme (CASP), containing ten questions to help confirm the validity of the research design and the originality of the results in comparison with similar studies. A series of inclusion and exclusion criteria were applied after the search, and 35 eligible articles from ten African countries with evidence of sexual violence, disclosure patterns, and APAC in camp settings were included in the study. Results described situations of SV in humanitarian settings in Africa as “terrible”, “bad”, “an epidemic”, and “severe” as girls were used as sex objects, for profile enhancement and as a weapon of war. We also found that the illegality of APAC in Africa is causing a high occurrence of clandestine abortions in conflict contexts. Disclosing SV among IDPs in Africa did not follow a clear-cut pattern but was generally determined by socio-demographic characteristics. Sexual health is a fundamental right of all, as enshrined in SDG 3, which makes this topic a major public health issue. We therefore conclude that although disclosure may aggravate stigmatisation in some instances due to adverse reactions, it is still crucial to the healing processes.

## 1. Introduction

At the end of 2022, the latest year for which data were summarised as of May 2023, global forced displacement hit a record high as the number of people displaced by war, violent conflicts, persecution and human rights abuses stood at 108.4 million, a 2.1% increase from the previous year [1]. Although violent conflicts have been almost endemic in some areas of Ethiopia, Afghanistan, Yemen, Congo DRC, Nigeria and Syria for over a decade now, the Russian invasion of Ukraine has worsened the situation of global displacement because more than 12 million Ukrainians have been displaced since the Russian invasion, adding to the high figures humanitarian organisations were earlier trying to grapple with [2,3]. This situation implies that compared with the present global population of approximately 8.1 billion people [3], one in every 14 people globally is displaced. Since the speed and volume of displacement are currently outpacing the availability of sustainable solutions such as reintegration, resettlement or local integration, accommodating IDPs temporarily in makeshift camps appears to be the cheapest solution, accounting for the thousands of IDP camps globally. 

The literature has established that women and girls constitute the majority of the displaced persons in camp settings [4,5]. Sexual violence and unwanted pregnancies are typical in displaced people’s camps (DPCs), and therefore pregnancy termination, in most cases, is prevalent among the affected [4,6,7,8]. Recent data show that about 56% of all unintended pregnancies globally were aborted, and estimates show that about 25 million unsafe abortions take place annually, the majority of them in developing countries [9]. Even though existing estimates show that complications due to clandestine abortions cause 25–50% of maternal deaths in refugee camps, the actual prevalence of abortion and its associated outcomes in humanitarian settings, both of which are assumed to be worse, remain grossly undocumented [10,11]. While the need for abortion services usually increases during humanitarian crises, the actual abortion needs and experiences of women in IDP camps are mostly ignored, which affects having good statistics on the prevalence of abortion in camps. Thus, reproductive healthcare, like abortion and post-abortion care (APAC), is essential in IDP camps, just like in any other society, so it will be easy to estimate the prevalence of abortion in humanitarian settings in the future. This service is necessary for two main reasons, among others. One, previous studies show that IDPs experience extremely high levels of sexual violence [4,9]. Two, displaced young girls and women are more likely to engage in transactional sex as a means of economic survival because, in most cases, their means of livelihood were destroyed before they were brought to camp [12,13,14]. As a result, compared with the general public, displaced persons may experience higher unintended pregnancy frequency, thus requiring a greater need for APAC.

Abortion and post-abortion care (APAC) is, however, not legalised or publicly approved in most sub-Saharan African (SSA) countries [15], including those hosting millions of displaced persons like Uganda, Sudan, Ethiopia and Nigeria. In Sudan, for instance, abortion was not legally approved during the civil war as various military movements encouraged pronatalist ideas for military gain to replace men lost to war [16]. Some of the barriers preventing women and girls from accessing APAC, particularly in developing countries and in camp settings, include the unavailability of abortion services in rural settings [17], providers’ fear of prosecution [18], cost [8], lack of knowledge about available services and stigmatisation [19]. These services are not publicly available for African refugees and displaced persons. However, the available evidence about access to sexual and reproductive health (SRH) in a humanitarian setting refers to post-abortion care (PAC) and menstruation regulation (MR) services made available for the Rohingya refugees in Bangladesh without any intimidation, stigmatisation or discrimination [7]. 

Sexual violence against women and girls in humanitarian settings is beginning to gain prevalence in the literature [20,21,22]. However, disclosing such an occurrence, especially in these settings, has proved difficult for the offended party. It is even more difficult if sexual violence in camp settings results in unwanted pregnancy because APAC is not provided in camp settings [4,6]. This situation is why many cases of unwanted pregnancy go unreported in these settings because, on most occasions, the offended do not know how to go about disclosure or to whom disclosure should be made in order to obtain justice. Overall, there is a need for more evidence of sexual violence in the camps, for APAC, as well as for an understanding of disclosure patterns. This step is necessary for policy formulation or adjustment, particularly within humanitarian settings in Africa. Moreover, it is important to specify that sexual and reproductive health (SRH) is a fundamental human right according to WHO and as shown in previous related studies [23,24]. Guidance and protocols for essential SRH services in humanitarian settings are also available in the literature [24]. There are also guideline interventions documented for mothers in humanitarian settings on pregnancy and lactation, women of reproductive age, adolescents, newborn babies and women with disabilities. Moreover, it is clearly enshrined in SDG 3 to ensure healthy lives and promote well-being for all at all ages by 2030. Specifically, Sub-section 3.3 of SDG 3 clearly admonishes all governments to end the epidemics of AIDS, …, and combat hepatitis and other communicable diseases by 2030. Sub-section 3.7 also admonishes governments all over the world to ensure universal access to sexual and reproductive healthcare services, including for family planning, information and education, and the integration of reproductive health into national strategies and programmes by the year 2030. How then can sexually transmitted infections like HIV/AIDS and hepatitis B be stopped from spreading if the epidemic of sexual violence is not stopped, especially among the vulnerable? The subject under discussion is therefore a major public health issue as it commands global attention as enshrined in SDG 3. Therefore, this study aims to synthesise and bring forward collective evidence on the topics of violence, disclosure patterns, and abortion and post-abortion care services in displaced people’s camps in Africa through an extensive scoping of existing studies. This will strengthen the collective knowledge base on the prevalence of SV in IDP camps and APAC, as well as how displaced persons have been managing disclosure of SV within IDP camps in Africa.

## 2. Abortion and Post-Abortion Care Services (APACS) among IDPs in the African Context

Abortion, unless to save the life of a woman, is severely restricted in most developing countries. This situation makes women who are required to terminate a pregnancy mostly resort to clandestine abortions, to which have been attributed 31% of maternal deaths annually [8,9]. Even in Uganda, considered a ‘safe-haven’ for IDPs in Africa because they are allowed to work and earn income [4], women are still not able to freely access abortion care due to legal restrictions. A study in some medium- and low-income countries [9] shows that only 7 out of every 1000 women who require abortions are treated in standard health facilities. The situation is likely to be worse in humanitarian settings in Africa, where health facilities, in most cases, are grossly inadequate. Even though the literature is clear about clandestine abortions in camp settings, data on the actual level of prevalence are grossly undocumented. It is still not clear from the available literature what percentage of abortions successfully performed in humanitarian settings, if any, were carried out in standard health facilities. The availability of such data will be a significant breakthrough towards finding a lasting solution to the situation of clandestine abortions in camp settings in Africa.

Recent studies show that due to a single or a combination of factors, such as lack of information about the legality or availability of safe abortion services in their new environment, fear of stigmatisation in the camp or their host community, financial incapacitation and inability to communicate effectively with care providers due to language barriers [14,25,26], clandestine abortion is rampant in IDPs camps. For example, fear of the perceived legal implications of induced abortion among Congolese refugees in Uganda makes them engage in clandestine abortion practices, such as using detergents, herbs, crushed bottles and large doses of oral contraceptives [9]. These challenges are likely to lead to a high prevalence of maternal deaths among IDPs, as estimated figures from the United Nations Population Fund [UNFPA] show that 25–50% of maternal deaths in IDP camps are due to complications from unsafe abortion [4,10,11]. Despite the severity of this problem, studies about the prevalence of SV and unsafe abortion in IDP camps, especially in Africa, have just begun to grow in the last decade. However, knowledge about displaced women’s abortion knowledge, attitudes and practices (KAP) in Africa remains scanty. There is a need for synthesised evidence on the subject in order to have sufficient science-based information that will guide the framing of appropriate policies on APAC, particularly as it relates to IDPs across Africa, which led to the objective of this study.

Furthermore, research on disclosure patterns among sexually violated women at IDP camps in Africa, especially when APAC is required, has just begun to gain attention in the last decade. A previous study among displaced women in North-East Nigeria shows that fear of stigmatisation may prevent those carrying pregnancies which occurred as outcomes of sexual violence from disclosing the pregnancy, causing late disclosure and affecting whether anyone is even informed at the end [27]. There is leadership and hierarchy in every society, and displaced people’s camps are no exception. In Nigeria, for instance, IDP camps are either controlled and managed by the government and humanitarian organisations or jointly managed and financed by both [12]. Whichever way, there should be a channel of disclosure where women and girls who experience sexual violence can disclose their situation without fear of intimidation or stigmatisation. Therefore, creating an enabling environment for disclosure about sexual violence, handled in a mature, anonymous and non-stigmatising way, would go a long way to reducing the post-traumatic stress disorder (PTSD) of the violated. It will also give them assurances of safety and access to APAC, should the need arise. There is a need for the assemblage of evidence from previous studies in Africa to come up with sufficient evidence, without which the epidemics of SV at IDP camps in Africa may be difficult to address. It will also go a long way towards adjusting policy, so that IDPs in Africa can have access to high-quality reproductive health services that meet their needs as a people.

Given the gaps in the literature, a scoping review of relevant articles was conducted, and the following three primary research questions were asked and answered. One, how prevalent is sexual violence at IDP camps in Africa? Two, what provision is made for abortion and post-abortion care (APAC) across IDP camps in Africa? And three, what is the disclosure pattern of the experience of sexual violence among violated women in the IDP camps? Finding accurate answers to these questions will help safeguard the health and dignity of African girls and young women in various IDP camps across Africa.

## 3. Methodology

### 3.1. Research Design

The study is a scoping review of relevant literature from African settings. In line with the review questions, the scoping review was conducted based on three thematic areas: sexual violence in IDP camps, disclosure pattern of sexual violence among IDPs, and APAC in IDP camps. The articles served to establish whether there are facilities for safe abortion as well as establishing a pattern of disclosure in case of sexual violence among IDPs in their camps in Africa. Girls and young women in humanitarian settings are the primary subjects of this study. 

### 3.2. Sources of Data

A systematic search of five databases, namely MEDLINE, PubMed, Scopus, Embase and Google Scholar, was conducted, and solely articles written in English were extracted for this research. These databases have some of the largest assemblages of population and public health and social science articles globally. Relevant grey literature was retrieved from the websites of humanitarian organisations such as the United Nations Population Fund (UNFPA), United Nations High Commissioner for Refugees (UNHCR) and International Organization for Migration (IOM), which are some of the relevant organisations on the subject of interest. Search terms such as sexual violence, IDPs and Africa, post-abortion care and IDPs, sex and disclosure were adopted to extract relevant articles. All studies retrieved and retained were conducted within the last ten years (between 2013 and 2023) so as to be abreast of recent updates on the subject matter. The retrieval and review process started in June 2022, was repeated in August and continued till January 2023, making six months of extensive literature search and review. 

### 3.3. Inclusion and Exclusion Criteria

While the search from PubMed produced 440 records, Google Scholar produced 492 records, Scopus produced 760 records, MEDLINE produced 680 and Embase produced 498. Collectively, the search engines produced 2870 records, of which approximately 31% were duplicates. The stages involved in the selection, inclusion and systematic review of the retrieved articles were guided by the PRISMA guidelines for systematic reviews (see Flowchart in Figure 1), as adopted by Page et al. (2021) [28]. The reason is that the PRISMA requirements are scientifically sound for a scoping review compared to an a systematic review. The Critical Appraisal Skills Programme (CASP), which contains ten questions to help cross-check the validity of the research design and the originality of the results compared to similar studies, was adopted. Furthermore, the protocol for this systematic review was also registered and published with the Open Science Forum (OSF) to ensure transparency of the research process. The protocol registration was initiated on 7 May 2024 and updated on 20 June 2024 (see https://osf.io/mu9hb/ (accessed on 4 April 2024), for further information).

In the first screening stage, about 1790 records were retained and further screened after we had excluded 884 and another 196 records for duplication and incompleteness, respectively. In the second screening stage, 948 records were retained after excluding 842 records as the authors performed title and abstract screening. Another 576 articles were removed after initial scanning, where unwanted articles were excluded, as shown in Figure 1. In the penultimate stage, authors became increasingly familiar with the remaining full articles, which led to the setting of further eligibility criteria. At this stage, we were left with 372 full articles for assessment, after which 201 were excluded because they were not conducted in Africa. Another 21 articles were removed because their methodologies were unsuitable, while some lacked a methodology altogether. Some include unrelated reports, technical papers, articles not published in English, incomplete works in progress, and those without sound backgrounds and genuine data sources. Another 115 articles were removed because of their irrelevant target population. Even though such articles were good and written well, the population was not IDPs; other women were not in a conflict context, which is outside the scope of this study. Finally, 35 full articles were retained and utilised because they met the eligibility criteria for inclusion in this research, as seen in Figure 1. They were studies conducted on IDPs, particularly in the camp context, studies whose authors were based outside of Africa but had the data gathered in Africa, had sound methodologies, used representative sample sizes and provided confirmable sources of data.

### 3.4. Ethical Consideration

Even though this study did not involve fieldwork to gather primary data since it was a scoping review, we deemed it necessary to seek approval exception due to the sensitivity of the subject matter and since humans were the object of the study. Therefore, we applied for an ethical approval exception from the Covenant University Research and Ethics Committee (see Appendix A).

### 3.5. Synthesis and Analysis Plan

Our synthesis and analysis plan follows that of Tricco et al. (2016) [29] on conducting and reporting scoping reviews. The framework includes the following five steps. One, identifying the research question by clarifying and linking the purpose and research questions. This step was completed by the corresponding author, P.O.A., but was later edited by the second author, S.A.A., who compared research questions with objectives to ensure objective accuracy. Two, relevant studies can be identified by balancing feasibility with breadth and comprehensiveness. This selection was made following Tricco’s third procedure for writing scoping reviews, which is the selection of studies using an iterative team approach to study selection and data extraction. P.O.A. and S.A.A jointly undertook this. The fourth procedure was charting the articles (see Figure 1) incorporating qualitative thematic analysis. The synthesis was purely qualitative (content analysis) for the research. This was achieved by organising and undertaking content analysis of all information using a systematic interpretation of words and their meanings, along with the study objectives. This information was later synthesised into sub-groups based on pre-determined themes, and thematic analysis was used to present the findings. The fifth procedure was to collate, summarise and report the results, including policy, practice and research implications. Both authors jointly executed this task. 

## 4. Results

From our reviews, fifty facility-based studies were included; that is, the study revolves around fifty IDP camps or humanitarian settings across ten countries in Africa. This implies that one or more of our selected studies used more than one IDP camp. The ten countries are Burkina Faso, Central African Republic (CAR), Congo DRC, Ethiopia, Kenya, Nigeria, Rwanda, Sudan, South Sudan and Uganda. These countries cut across Central, East, West and North Africa, and they host some of the largest IDP camps not just in Africa but in the world, especially Uganda, Sudan and Nigeria. The total sample size of all the 35 included studies was fifty-seven thousand, seven hundred and sixteen (57,716) IDPs. Many of the studies adopted qualitative techniques such as focus group discussions (FGDs) and key in-depth interviews (KIIs), while others adopted quantitative techniques making use of structured and open-ended questionnaires. The articles reviewed were all published within the last decade (i.e., 2013 to 2023).

The three sub-themes used for the presentation of results and analysis were sexual violence in displaced people’s camps in Africa, abortion and post-abortion care (APAC) in displaced people’s camps in Africa and disclosure pattern of sexual violence among the displaced in Africa. Having synthesised the articles reviewed, the results were thematically analysed as presented below. Studies on the situation of sexual violence, disclosure patterns and APAC adopted for this study have also been summarised in Table 1.

### 4.1. Sexual Violence in Displaced People’s Camps in Africa

The nexus between forced displacement and sexual violence (SV) and their attendant consequences for adolescent girls and young women in humanitarian settings are documented [6,9,36,40,41,45,46,47,48]. Several African-based studies on displaced persons in the camp context have exposed how women and girls were used as mere sex objects and means of achieving goals with little or no respect for opinion or consent [8,18,30,31,33,39,49]. The description of SV at IDP camps across Africa is depicted in Figure 2 and presented in Table 2 as well. 

In northern Nigeria, for instance, a study in an IDP camp [30] showed that the categories of women that were most affected were adolescents, unmarried women, uneducated women and women of Hausa extraction. Another empirical study in humanitarian settings in north-eastern Nigeria showed that many young IDPs (girls and women) were forced to engage in transactional sex or child marriage as a means of livelihood due to the lack of essential basic needs in their camps [41]. This caused several unwanted pregnancies and clandestine abortions due to the low level of contraception in their camp. Still in Nigeria, a qualitative study undertaken between 2018 and 2019 among conflict-induced IDPs temporarily accommodated in various makeshift camps in Abuja, noted that bodies of female IDPs were “objectified” and used for sexual gratification, procreation, profile enhancement, care-giving and as weapons of suicide bombing by Boko Haram [31]. In a related study by Obiefule and Adams [39], female IDPs in various camps in North-East Nigeria faced challenges such as sexual abuse, human rights violations, starvation, and social ostracism, among others. Sexual violence is said to be the “most pronounced” [9,30,36,39] among the various challenges female IDPs face in camps across the continent, as seen in Figure 2. It is even believed that violence against IDPs is underreported in Nigeria [30] despite many studies on it in the last two decades. Although a study [36] noted that security measures were put in place by camp officials to mitigate sexual violence, the strategies were, however, inadequate as several incidents still went unnoticed and unreported. Sexual violence against female IDPs in Nigeria was described as “severe” in a study conducted among 403 randomly selected IDPs [33] and also made achieving quality education an aspiration rather than a reality.

The situation of IDPs in camps in Rwanda is very similar to those in Nigeria. In a study among Congolese refugees in Rwanda [36], rape, unwanted physical touching, sexual exploitation, transactional sex, child marriage and girl trafficking were the key forms of sexual violence experienced by girls in a particular camp. This was perpetuated due to the difficult life in camp and poor security surveillance in this camp. In another empirical study by Njoku and Akintayo [46], in-depth interviews with executives and programme officers of NGOs in conflict and humanitarian settings corroborated ongoing discussion on sexual violence against girls in north-eastern Nigeria. This study found that the harsh economic situation of women in camps forced them into transactional sex in exchange for livelihoods despite the associated socio-legal risks involved.

The situation is similar in the Central African Republic (CAR) where the SV situation among displaced girls and women was described as “an epidemic” due to security fragility resulting in serious displacements in the country [50]. The situation of SV among displaced girls and women is similar in Uganda where the level of prevalence was reported to be high, to the extent that one in every three women confirmed being a victim, including of child marriage and rape [37]. Many women in northern Uganda who experienced SV in the past, according to the findings of Murphy et al. [37], are reported to still have physical and psychological trauma, untreated sexually transmitted infections and depression.

### 4.2. Abortion and Post-Abortion Care (APAC) in Displaced People’s Camps in Africa

Clandestine abortion among displaced persons is partly due to the lack of access to APAC in Africa as a result of the restrictive laws regarding induced abortion [51]. According to Ikenye [51], despite the ratification of many treaties (both national and international) on safe abortion in Nigeria, implementation remains poor [51]. In an empirical study in a conflict context in north-eastern Nigeria, all women who became pregnant as the outcome of sexual violence requested that they be terminated. However, such requests were declined because abortion services were not offered in the clinic because of Nigeria’s restrictive abortion laws [6]. In Uganda, abortion is legally restricted among the general public, let alone IDPs. Uganda houses the highest number of IDPs in Africa [9], so the need for APAC is heavy, yet it is highly restricted. Uganda hosts 1.4 million conflict-induced IDPs, making it the host of the highest number of these vulnerable persons in Africa [4]. Although Uganda is reported to be the haven of IDPs in Africa because IDPs are supported in securing a means of livelihood, yet displaced women and children repeatedly have unmet needs for sexual and reproductive health there as abortion is legally restricted [4]. Belief about abortion is largely negative in Uganda, irrespective of the circumstances surrounding it. In empirical research [9], the majority (>90%) of IDPs avoided those who terminated pregnancies and perceived them as “bad girls”.

In a study by Casey et al. [26] in 62 different IDP camps and health facilities across Burkina Faso, Congo DRC and South Sudan, light reproductive health services such as child delivery services and sexual health counselling for young women were available in some of the camps. While three facilities in DRC provided selective treatment for rape victims, none of the camps or health facilities across the three countries made provision for safe abortions. A multi-method qualitative study on maternal health and delivery care, contraception, APAC services and their intersection with sexual violence and GBV was recently undertaken among Congolese refugees in Uganda. Results show that Congolese refugees in Uganda are unable to find their way around legal restrictions on abortion, resulting in the proliferation of clandestine abortions among them, not just among those in the camps but also those who stay in host communities [4]. In the case of South Sudan, for instance, discussions about unintended pregnancy and abortion are viewed as “taboo”, thus exacerbating poor sexual and reproductive health outcomes among refugees and the IDPs [8].

If it is legally difficult to access proper abortion care, then accessing post-abortion care too is likely to be a hassle. Humanitarian organisations which try to secure APAC for victims in conflict settings face legal, operational, cultural and political challenges [42,52]. Generally, there is a negative perception of induced abortion in Africa. Unfortunately, the need for it appears to increase during conflict and displacement due to increases in transactional sex, disruption of marital vows, rape and low contraceptive usage [8].

### 4.3. Disclosure Pattern of Sexual Violence among the Displaced in Africa

Disclosure of sexual violence has been regarded as a social experience that may support the survivors or affected persons towards healing or aggravate stigmatisation depending on who is handling it and to whom the disclosure is made [53]. Mostly, survivors of sexual violence are hesitant to disclose the assault or seek services or resources for fear of the news spreading to others in the camp [48]. Current evidence suggests that disclosing sexual violence is not the main challenge for the victim but the post-disclosure reactions of the hearers [53]. An empirical study among survivors of sexual violence showed mixed reactions among formal and informal groups after disclosure (negative and positive reactions). The study concluded that although the victim may experience negative reactions, disclosures are, however, crucial to the healing processes [53].

Focus group discussions (FGDs) and in-depth interviews (IdIs) were conducted among Congolese refugees about SV and abortion experience. The disclosure pattern was indirect, as most IDPs who have experienced abortion in the past do not talk about their experience. Rather, they prefer to share the experience of other IDPs, perhaps a close friend, particularly those who used clandestine means to undertake the abortion [9]. This “talk about others” attitude helped researchers to gather that abortion practices among Congolese refugees are predominantly through clandestine means [9]. Other similar empirical studies among IDPs in Congo and Ethiopia revealed that displaced persons prefer “group-based” disclosure of sexual violence when dealing with humanitarian workers to reporting to individuals [48]. The study suggests that a group-based qualitative study offers greater promise in terms of generating detailed information that could help policy formation or adjustment than individual interviews.

In an empirical study undertaken among IDPs in Abuja, north-central Nigeria, disclosing cases of sexual violence among IDPs, sometimes termed stress disclosure by Alhassan, Akuki and Hezekiah [32], was very difficult. IDPs preferred to keep such to themselves and continue to suffer emotional stress rather than inform camp officials who they felt might ‘broadcast’ the situation and aggravate their pain through stigmatisation. In another empirical study by Oladeji et al. [6], most women who experienced sexual violence-related pregnancies (SVRPs) in IDP camps in north-eastern Nigeria first disclosed the pregnancy to their peers before disclosure to healthcare providers or family members. About 60% of such women were said to have left the camp, and the outcomes of such pregnancies remained unknown when they did not have access to abortion services in the camp. In CAR, an empirical and nationwide study of 25,143 displaced adults showed that disclosure of SV was on the increase, but the pace of progress was still very slow [50]. One in every five victims was found to have felt safe to file a formal complaint with security and stakeholders in the judicial system in CAR. Fear of retaliation, re-traumatisation and stigmatisation were identified as factors causing the slow rate of disclosure of SV in CAR. The situation is no better in Uganda, as a study [37] revealed that women who experienced SV were not able to freely disclose the situation because of fear of aggravated stigmatisation. The only point of succour was moral support from co-survivors.

Reporting of SV among violated girls and women refugees in South Sudan follows the same pattern as other countries in terms of pace but is a little different in terms of reporting. Generally, survivors show apathy towards disclosure or seeking help from camp officials [40]. However, it was also found that SV occurred at two levels: non-partner SV, from people the women did not have aromantic relationship with prior to the incident, and that from intimate partners, but these two did not show similar reporting scales. Those violated by non-partners were more likely to disclose their experience than those where SV was perpetrated by intimate partners [47]. Moreover, disclosure and help-seeking behaviour of the violated were affected by certain socio-demographic factors such as age, perpetrator’s identity, working status of the woman, poverty and location of occurrence [47]. However, for those violated by partners, the severity and longevity of the violence determined whether they would disclose their experience. If the event persisted, they were likely to report it but they hardly ever sought health services post-occurrence, as shown in this study, compared with those suffering non-partner SV [47]. Women who have experienced SV in the past have difficulty in their relationships with others as they face stigmatisation and systematic snubbing from friends and family. This is also extended to children born in such situations in humanitarian settings [37]. It is important to note that some of these studies have been summarised in Table 2.

## 5. Discussion

First, it is important to specify that sexual and reproductive health (SRH) are fundamental human rights according to WHO and as shown in previous related studies [23,24]. The guidance and protocols for essential SRH services in humanitarian settings are also available in the literature [24,54]. Existing guidance, however, is more focused on service delivery related to childbearing in humanitarian settings. There are guideline interventions documented for mothers in humanitarian settings on pregnancy and lactation, women of reproductive age, adolescents, newborn babies and women with disabilities [54,55]. However, the task of protecting the sexual health and rights of people in humanitarian settings has not been given its rightful place despite their being very sensitive public health issues. The United Nations has not shied away from the importance of the sexual rights of people, which is why issues on sexuality stay as number 3 out of the 17 SDGs listed to be fulfilled by 2030. Unfortunately, none of the above services include guidelines for women to terminate a pregnancy, even if it is unwanted. This is despite the fact that SV is rampant in humanitarian settings in Africa. The literature describes the situation as bad in Nigeria, severe in Rwanda and as an epidemic in CAR [9,30,36,50]. There are no statistical data to quantify the level, but, from the reviewed studies, the situation is severe. It is common to describe the causal factors as lack of means of livelihood among young girls in the camp or other socio-cultural factors, but one thing is certain: SV is endemic in humanitarian settings in Africa. However, it is important to note that SV against women and girls in IDP camps is not limited to Africa. Indeed, similar situations have been documented among vulnerable groups in Asia, the Middle East and South America, particularly Colombia [45]. Females were found to be more susceptible to SV in these settings, and one in four forced marriages took place before the girls reached 18 years of age. For instance, drawing examples from Colombia, a country which has the highest population of IDPs in the world [56], is important. After about 50 years of civil war, Colombia is home to 7.7 million IDPs. Despite a very strong Christian faith, where the majority of the population of Colombia belong to one Christian denomination or another (79% Catholic and 13% Protestant), IDPs still face a high risk of SV [56]. In Asia, the case of Rohingya girls is still very prominent in the literature. A study [57] confirmed that the vulnerability level of Rohingya women and girls is high because of their gender, refugee status and ethnic affiliation. Even before they arrived at Cox’s Bazar camp in Bangladesh, they were subjected to violent assaults by the Myanmar Army in Rakhine State before they fled. Unfortunately, SV against them worsened in the camp, partly due to the breakdown of family and community structures [57]. At Cox’s Bazar camp, this study [57] revealed that beliefs, norms, attitudes and structures that promote male–female unequal power relationships worsened the situation of gender-based violence. Rohingya women and girls went through serious rights violations in Myanmar before they were finally forced out in 2017. Many anti-Rohingya policies and the attacks of 2017 led to the rape and murder of many Rohingya women [58]. Those who were forced out were eventually hosted in Cox’s Bazar District in Bangladesh. Sexual violence in humanitarian settings does not respect geography, ethnicity or skin colour. What qualifies adolescent girls and young women for exposure is being displaced, a situation which aggravates the likelihood because opportunities for self-defence and rights protection seem limited. The majority of adolescent girls and young women interviewed in a recent study [59] had experienced SV in one form or another. Cases of unacceptable sexual touches, intimate partner violence (IPV) and outright rape were recorded. Syrian refugee girls and young women in Lebanon are not exempted from this challenge, as shown in a recent empirical study [60]. So, the challenge of SV in conflict settings goes beyond colour and race; it is global in scope and dimension.

Abortion and post-abortion care (APAC) are hardly made available in humanitarian settings in LMICs. It is surprising that even in Uganda, which hosts the highest number of displaced persons in Africa, such a provision is not made. Despite several agreements which have been signed by African countries on abortion legalisation, most have yet to ratify those agreements. Of all the IDP camps in Africa, only in Sudan and Uganda are light reproductive health services, like delivery services and menses control services, provided [7,26]. Contraceptive counselling is allowed in humanitarian settings in Nigeria and CAR [23]. It is, however, heart-warming to note that post-abortion care (PAC) is on the increase across IDP camps in Africa and even beyond. For instance, recent studies undertaken in humanitarian settings in Congo DRC, Somalia, Uganda and Yemen [23,24,26,60] have proved that post-abortion care exists and that demand is even on the increase. The question still begging for an answer is where the abortions are taking place. Are they taking place in the camps or elsewhere? Are they performed professionally, handled by experts, or clandestinely? The picture painted here aims to clarify that although there are health facilities in IDP camps, they are mostly not adequate compared with the number of women and scale of services demanded. It is also noteworthy that the range of services is limited, especially in sexual and reproductive health. For instance, in Nigeria, most IDP camps have at least a small health facility providing malaria treatment, treatment for accident victims and other services, but APAC is strictly prohibited. However, for IDPs in Africa, the story is not the same as for their counterparts in Asia. For instance, there is provision of sexual and reproductive health services for the Rohingya women and girls in Cox’s Bazar District in Bangladesh, where they are hosted. Although the service provision is grossly inadequate according to a study [57], however, services are available, unlike in Africa where provision does not exist. It was also found that although the provision of sexual and reproductive health is available in Bangladesh, socio-cultural factors among Rohingya have contributed to limiting the provision of reproductive health services in the camps. So, it is not a case of lack of provision but a case of limited utilisation because of socio-cultural beliefs.

Disclosing sexual violence against a girl or woman to another person is never easy. Studies have shown that the violated fear shame, stigmatisation and post-disclosure attacks by perpetrators if they disclose what transpired. This is why a recent study confirmed that there is an inverse relationship between the socio-economic status of the perpetrators of SV and disclosure by the victim. If the person who perpetrated the SV is a bigwig in the camp or someone very influential, victims may not want to disclose the issue for fear of being attacked again. That is why many victims of SV encounter mental health challenges because they are ‘bottling up’ a lot. It is also important to note that studies have shown that the mental health and psychosocial consequences of sexual violence against male and female survivors may be similar, but the way each processes trauma displays symptoms, seeks help, adheres to treatment and improves their mental health differs by gender [61]. Women and girls display graver symptoms of SV than their male counterparts, which is why women and girls need to be protected. For instance, between September 2017, when the Rohingya girls and women were forced out, and August 2018, 52 maternal deaths occurred in Ukhia and Teknaf Upazila camps in Cox’s Bazar District [58]. Why was this so? It may partly have been caused by the dominance of clandestine abortion.

## 6. Conclusions, Recommendations and Suggestions for Further Studies

This review provides an evidence synthesis from the available literature on sexual violence, disclosure patterns and APAC in conflict settings to inform potential strategies for interventions and also to tease out priority areas for further research. It has been found that SV is still rampant at IDP camps across Africa and is grossly underreported. It was also found that APAC is rarely provided at IDP camps in Africa, unlike the situation at the camps in Asia and other parts of the world. It was also established that the fear of aggravating stigmatisation tends to prevent a lot of survivors from disclosing their cases at IDP camps in Africa, causing underreporting of the real figures or number of occurrences. Finally, we conclude that even though disclosure may aggravate stigmatisation in certain cases due to negative reactions, it is, however, still crucial to healing processes.

Therefore, more surveillance must be mounted at IDP camps across Africa whereby perpetrators can be caught, no matter how secretive they may be. If security is tightened up and all loose ends tied, cases of rape may drop in humanitarian settings in Africa. Moreover, African countries need to ratify agreements made for legalising abortion in their countries. This is especially important for the ten countries under study so that unwanted pregnancies, especially those arising from SV, can be freely terminated. For those who are violated, disclosing SV is a step in the right direction towards healing. Therefore, mechanisms should be put in place to encourage any violated IDP to disclose the issue to access necessary help. More importantly, there is a need to train health workers on positive attitudinal changes towards promoting a rights-based, fear-free, non-judgmental, and non-discriminatory approach to girls and women who seek APAC in conflict settings.

The whole essence of this research is to ensure that SV is dealt with in humanitarian settings and that those who are violated obtain justice. However, the most glaring gap in the literature is the unavailability of data or statistics on the proportion of abortions that take place in humanitarian settings compared to what happens in the wider society. Statistics on the number of abortions that take place in humanitarian settings may help to provide evidence to advocate and pressure related institutions to improve women’s and girls’ care related to sexual and reproductive health and rights in these settings.

## Figures and Tables

**Figure 1 ijerph-21-01001-f001:**
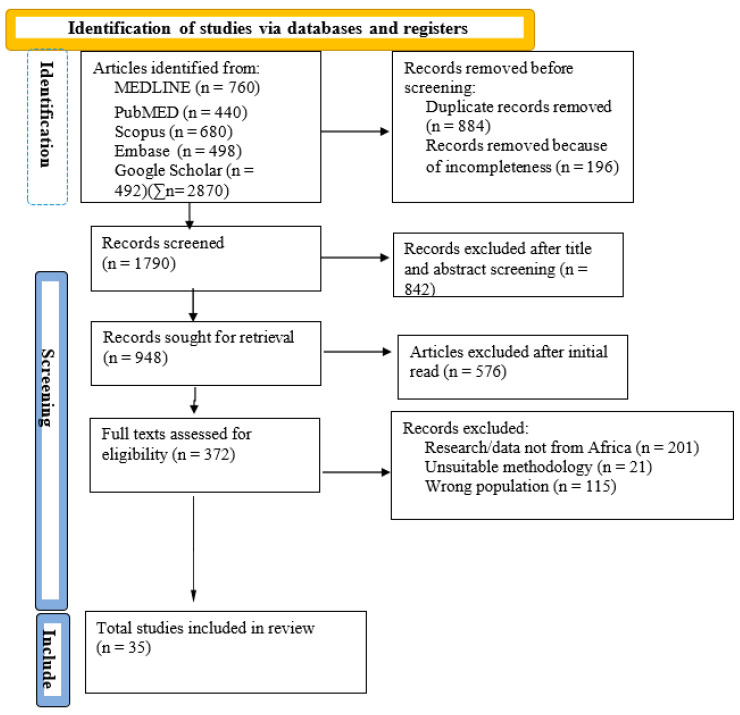
PRISMA Flowchart on Included and Excluded Records.

**Figure 2 ijerph-21-01001-f002:**
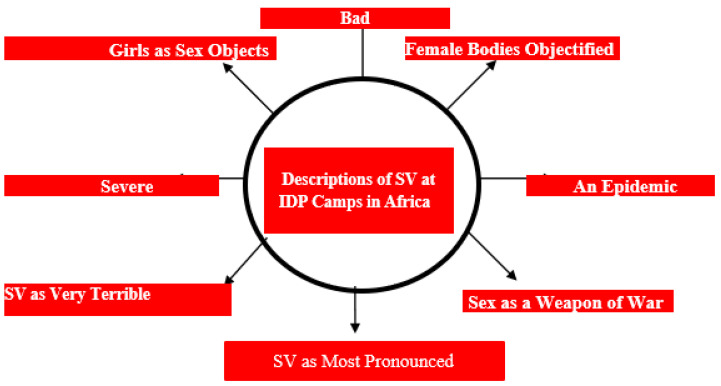
Descriptions of sexual violence situation at IDP camps in Africa, 2013–2023. Source: Authors, 2024.

**Table 1 ijerph-21-01001-t001:** Articles adopted for the study: selected.

Author	Study Purpose	Study Design	Country/Research Area	Study Results
Adejumo et al. (2021) [30]	To assess the GBV experience of IDP women in their camp	Quantitative cross-sectional study involving 288 women IDPs in Edo State, Nigeria	Edo State, Southern Nigeria	- SV was the most common post-conflict challenge faced by women IDPs - This mostly affected adolescents and unmarried women
Agbaje, F.I. (2020) [31]	To examine ways the Boko Haram group objectified the female body and factors fueling such action	Qualitative study adopting FGDs with 50 girls and IdIs with two officials at two IDP camps in Abuja, Nigeria	North-eastern Nigeria	- Female IDPs were used for sexual gratification and profile enhancement by militants - Female IDPs were also used for suicide bombing
Alhassan et al. (2019) [32]	To investigate the influence of the intensity of events, distress disclosure and resilience in post-traumatic stress disorder (PTSD) among IDPs in Abuja	Cross-sectional quantitative research design. A convenient sampling technique was adopted to randomly select 55 participants for the study	Abuja, Nigeria	- There was no significant relationship between distress disclosure and PTSD
Bawa et al. (2022) [33]	To examine the impact of violence and forced displacement on female education	Mixed methods: quantitative and qualitative research designs were adopted	North-eastern Nigeria	- Displacement caused sexual violence and disruptions to female education in Maiduguri
Bermudez et al. (2018) [34]	To examine the nature of violence against adolescents in Kiziba Camp, Rwanda	Qualitative study adopting FGDs	Rwanda	- Economic insecurity and resource constraints caused overcrowding in houses in the camp, and adolescents travelled far to collect firewood - Protection mechanism and reporting pathways were underutilised because adolescents expressed concern over stigmatisation arising from disclosure of violence in the camp
Erhardt-Ohren and Lewinger (2020) [9]	To gauge the knowledge of refugees and IDPs on abortion, attitude and practices in LMICs	Qualitative research design	Displaced persons’ camps in LMICs	- Knowledge of abortion was moderate but the practice was embarrassingly low among refugees and displaced persons in LMICs
Hossain et al. (2020) [35]	To examine the relationship between disability, experience of GBV and mental health among refugee women	Cross-sectional quantitative research design	Kenya	- Refugee women with disability experienced a higher proportion of mental health challenges (anxiety, PTSD and depression) arising from SV, IPV and NPSV than those without disability before and after arriving in the camp
Iyakaremye and Mukagatare (2016) [36]	To examine the nexus between forced migration and sexual abuse, particularly among adolescent girls	Qualitative research design using FGDs and IdIs	Kikeme Refugee Camp, Rwanda	- Forms of sexual violence in the camp included rape, unwanted physical touching, sexual exploitation, commercial sex, early marriage and girl trafficking - These negatively affected the girls’ social integration and mental as well as reproductive health
Murphy et al. (2020) [37]	To examine disclosure and help-seeking behaviours of survivors of SV	Cross-sectional quantitative survey	South Sudan	- The odds of reporting SV in conflict settings was higher than in non-conflict settings - Socio-economic status of perpetrators affected the odds of reporting
Murray et al. (2021) [38]	To provide valid measures for sexual violence stigma for incorporation into monitoring and evaluation programmes in humanitarian settings	Cross-sectional quantitative research design	Somalia and Syria	- Stigmatisation from SV affected Somalian survivors differently from those in Syria, but both showed symptoms of depression evidenced by being withdrawn from others
Nara et al. (2019) [4]	To assess the reproductive health needs of displaced Congolese women in camps	Multi-method qualitative study	Uganda	- Congolese refugees in Uganda were unable to access abortion care, thereby resorting to clandestine abortion practices
Obiefuna and Adams (2021) [39]	To investigate the response of religious associations to humanitarian crises faced by female IDPs in NE Nigeria	Mixed methods: quantitative and qualitative research designs were adopted	Borno State, Nigeria	- Female IDPs in NE Nigeria faced SV, environmental racism, hunger and educational marginalisation - Religious associations’ response to the plight of female IDPs in NE Nigeria was rated 98.1% and commended but could be intensified
Oladeji et al. (2021) [6]	To report on the disclosure and outcomes of sexual violence-related pregnancies (SVRP)	Snowball technique was adopted to identify women with SVRP	North-eastern Nigeria	- SVRP was common in humanitarian settings, and fear of stigmatisation prevented victims from early disclosure - Women with SVRP preferred to disclose the situation to their close friends prior disclosure to healthcare providers or family members
Pham et al. (2020) [40]	To access the magnitude of SV and trust in the judicial system to prosecute offenders	Adopted a qualitative research design. Interviews were conducted in 4 waves among 25,143 adults resident in humanitarian settings in the Central African Republic between 2017 and 2018	Central African Republic (CAR)	- SV in conflict settings in displaced people’s camps was so rampant that it was described as “an epidemic” - Respondents positively perceived efforts to combat SV, and trust in the judicial system to prosecute offenders was growing - Disclosure was still, however, low as only one in five victims disclosed their ordeal
Stark et al. (2017) [41]	To compare disclosure behaviours of the sexually violated between individual and group reporting	A mixed methods research design was adopted for this study. Quantitative data were obtained from 1788 IDP girls and adolescents in Sudan and Ethiopia. For qualitative sources, data were obtained from 165 adolescent girls across 28 camps in Ethiopia and Sudan, comprising 5 or 6 girls in each group	Congo DRC and Ethiopia	- Group-based disclosure produced more information than individual interviews among the sexually violated
Tran et al. (2021) [42]	Capacity training to strengthen healthcare providers’ capacity to provide safe abortion and post-abortion care services in humanitarian settings	Mixed methods: quantitative and qualitative research design	Congo DRC, Nigeria and Uganda	- Participants’ knowledge on uterine evacuation and other abortion care was improved
Williams et al. (2018) [43]	To identify existing social and economic vulnerabilities of female adolescents in refugee camps in Rwanda	Qualitative study using FGDs and KIIs	Rwanda	- The convergence of material deprivation, lack of economic opportunity and vulnerability led to transactional sex and exploitation within and around the camps
Woldetsadik et al. (2022) [44]	To understand women’s perceptions of and experiences with conflict-related sexual violence (CRSV), especially the health and social challenges they constitute	Qualitative research design with the adoption of in-depth interviews (IdIs) to elicit data	Northern Uganda	- All women interviewed reported having experienced one form of SV or another, like rape and forced marriage - Many of the women could not relate well with friends and neighbours because of stigmatisation associated with the aftermath of rape

Note: SV: Sexual violence; IPV: Intimate partner violence; NPSV: Non-partner sexual violence; Congo DRC: Democratic Republic of the Congo; FGD: Focus group discussion; IdI: In-depth interview; KII: Key in-depth interview; GBV: Gender-based violence; PTSD: Post-traumatic stress disorder; LMICs: Low- and middle-income countries; SVRP: Sexual violence-related pregnancy.

**Table 2 ijerph-21-01001-t002:** Forms of Sexual Violence at IDPs Camps in Africa.

S/N	Forms of Sexual Violence	Country of Occurrence
1	Sexual exploitation	Congolese refugees in Rwanda
2	Transactional sex	North-eastern Nigeria
3	Child marriage	Nigeria/Rwanda/CAR
4	Girl trafficking	Congolese refugees in Rwanda
5	Sex objects	Central African Republic
6	Sexual gratification	Abuja, Nigeria
7	Sex for profile enhancement	North-eastern Nigeria
8	Unwanted physical touching	Congolese refugees in Rwanda
9	Rape	Rwanda, Uganda

Source: Authors’ compilation from reviewed articles, 2024; Prominent among them include: Iyakaremye and Mukagatare (2016) [4], Nara et al. (2019) [6], Tran et al. (2021) [36] and Oladejii et al. (2021) [42].

## Data Availability

The original contributions presented in the study are included in the article. Further inquiries can be directed to the corresponding author.

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
