# Peer review of "Sexual Violence, Disclosure Pattern, and Abortion and Post-Abortion Care Services in Displaced People’s Camps in Africa: A Scoping Review"

_ijerph, 2024, doi:10.3390/ijerph21081001_

Round 1
Reviewer 1 Report
Comments and Suggestions for Authors
Dear Authors,
Thank you for submitting your paper entitled “Sexual Violence, Disclosure Pattern, and Abortion and Post Abortion Care Services in Displaced Peoples’ Camps in Africa: A Scoping Review.” Your research topic is of significant importance as it delves into the existing studies on sexual violence (SV), disclosure patterns (DP), and abortion, including post-abortion care (APAC) within the context of humanitarian aid. The article is well-structured, addressing each point in detail in relation to the research questions, and maintains coherence with clear explanations throughout. However, there are minor revisions required for clarity and accuracy:
1. Table Sequence: The tables should be presented in the order of their reference within the text. Therefore, Table 2 and Table 1 need to be swapped to reflect their correct sequence.
2. Article Summary in Table 2: It is advisable to include the number of study samples in the summary of articles presented in Table 2. This additional information will provide a clearer picture of the scope of each study. The table headings on pages 8 to 10 (Author, Study Purpose, etc.) are also incorrectly placed. Kindly adjust these headings to their proper positions for better readability.
3. Reference Correction: On page 6, line 227, the citation ‘…that of Tricco et al. (58)’ is incomplete. It should be updated to…that of Tricco et al. (2016) to reflect the referenced work accurately.
Author Response
I have provided my response to all the issues raised by reviewer 1 in the following attached file. Please see the attachment
Thank you.

Reviewer 2 Report
Comments and Suggestions for Authors
Overall: This manuscript has potential to present a valuable and important collection of articles to demonstrate the current status of a topic of great importance. The reported process steps and presentations of data including visuals are contributory. Additional work is needed for this manuscript to communicate clearly, accurately, and in a non-biased way, the systematic findings that speak for themselves, versus argumentative or persuasive writing amidst the scientific report. All comments provided are intended to help strengthen the contribution of this work.
The focus areas/goals of the review would benefit from a more precise description of their connection/linkage … while SV disclosure and APAC care are important components and have a linkage, drawing this linkage more to why these were objectives of this review would strengthen its usefulness. Consider narrowing and further pinpointing linkage and intent in abstract, at line 97-98.
Monitor for biased and argumentative components vs. a scientific review (review across manuscript. Some examples: lines 116-120; line 247, “beautifully done”; lines 325, 329 describing situation as “bad” vs. reporting what is reported.
Highlights: Consider tightening language to more straight forward/academic reporting:
Pg 1, line 16: Recommend 2nd sentence towards “Although skeletal reproductive services are offered in some IDP camps in South Sudan and Uganda, main APAC services are highly prohibited.”
Recommend similar rewording for Pg 1, line 19 to be more objective: “Disclosure of sexual violence by the violated remains (consider reframe that gets more to specific summative finding)
Abstract:
Would add a descriptive/specific word to first sentence so it is clear on its own about context of review – ie “Violent political conflicts”?
Recommend modification at line 28/29 “… Scholar and determined ## articles retrieved met criteria for inclusion.”
Consider reword of sentence at lines 30/31 to be more summative and precise.
Introduction:
Check spacing/number intended on line 44
Consider removing “this is why” at start of sentence at line 101 to maintain formality of scientific approach.
Reconsider sentence at 104 for how to be more scientific and precise regarding the goal of this review – to synthesize and bring forward collective evidence on topic
Editing:
Check for division to two sentences or use of commas to improve clarity at line 64-65.
An overall screening for clarity and sentence structure, including comma use, is needed.
Recommend an overall screen for use of “that” – ensure it is only used when necessary to sentence structure to reduce distraction and smooth flow.
Comments on the Quality of English Language
Recommend a review of language use across manuscript to ensure clearest English meaning and intention. For ex (not inclusive, review across manuscript): pg 2/line 54 – “solution, accounting for the thousands of IDP camps globally.” Some items include “value” language that diminishes the objective, scientific intention of the review format (e.g. “seriously frowned upon” at line 80; “unfortunately” at various points to describe service availability in establishing background. Check sentence order to ensure clarity (e.g. line 87-88); line 110, sentence structure; line 212 “just random women” needs reconfiguring.
Reviewer 3 Report
Comments and Suggestions for Authors
Thank you for the opportunity to read this review. The review has a potential to contribute towards the understanding of sexual violence and reproductive health at the humanitarian settings. However, the way the review is written does not warrant publication. It needs an extensive revision. in particular, the methods and results section needs to be written clear and explicit which is deficient in the manuscript. Please find the feedback for the various secitons of the review.
Abstract – which quality appraisal tool was used? There is no information about number of included studies or study population in the abstract.
Introduction
Line 44 – confirm if it is a 21% increment? If yes, need some elaboration and critical reflection of this change
Line 58 – authors have used double negative wording – not recommended in academic writing
Introduction is very long and contain repetitive information. Need to be reviewed and re-structured.
Methods:
Search strategy needs to be provided as a supplementary material/document. Authors need to explain how they ensured reproducibility of search in google scholar.
Did authors register the review? If yes where, if no why not? Is the protocol published? Else how did authors ensure the transparency of the research process?
Section 3.3 - Inclusion criteria and exclusion criteria: This section does not describe either inclusion or exclusion criteria. Instead it describes flow of articles – which should be reported in the results section – not in the methodology. Besides there are a lot of flaws on what is written in this section which makes methodology of this scoping review very poor.
Line 199 – Need to clarify what incomplete records are? How did authors determine if a record is complete or incomplete without screening them? Please elaborate and justify
Line 199 – How did the authors identify duplicate records? The process has to be explained? How were citations managed? How were Google scholar records exported to citation tool? Which software was used to manage the review? And to which extent?
Line 200-202 needs to be clarified
Line 207 – Can you please explain and elaborate what authors refer by unsuitable methodology or no methodology? This needs further clarification in the inclusion and exclusion criteria.
Line 208 – As the authors have included grey literature, why are reports excluded during screening?
209 – what is the need for excluding papers without sound background – methodology is what is important
215 – how can study be conducted outside Africa but data gathered in Africa???
PRISMA Flowchart – the font used is not consistent. Please use consistent fonts
230 – “This was beautifully done by the corresponding author…” not an academic language. Are authors humblebragging themselves? It is prominent elsewhere in the paper. NOT GOOD RESEARCH PRACTICE.
Section 3.5 – this is a methodology section and again the authors are describing the results (line 233-238). The authors should focus on how data was synthesized (line 241-246), if there were any statistical analysis and how was risk of bias estimated? Do not need to describe whether it was beautifully done by author 1 or not? Line 245 does not make sense. Line 248-250 should go into results. 251-253 is humblebragging.
Results: Start with description of PRISMA flowchart and demographic information of the included studies. Line 255-56 should be dropped off… it is for readers to determine if anything is interesting. IF authors have reported what is interesting to them, then this is selective reporting of the results. Thee paper should be rejected.
Line 257 – said 50 facility-based studies – however the number of included studies was 35? Please explain the disparity.
A lot of repeated information in the paragraph 255-68. Please review.
Line 264 – was these many people interviewed? Were all included studies qualitative? Please describe the included studies - nature, sample size range, data collection procedures, general demographic information – male/female/both – this information are pivotal to the review.
Authors need to revise table 2 – 1. specially study design – Qualitative (elaborate type – FGD, interviews etc) – when telling mixed methods, no need to further say ‘quantitative and qualitative research design were adopted’ and 2. Sample size
Table 2 Stark et al. - Qualitative cross-sectional survey – please explain this research design – how can a survey be qualitative?
Pham et al. - Multistage sampling is not a study design – its just a sampling procedure; similar for Oladeji et al.
Table 2 – studies are arranged haphazardly. Please see APA format on how to arrange them – should arrange them in alphabetical order
Section 4.1 it is not clear if this section is a result or a discussion. It looks more like discussion than results. Need to revise the section. Please describe what you have found in your included studies. For example, which studies describe SV as female bodies objectified and how?
How can ‘severe’ describe SV? Please justify?
Why does table 1 come after table 2? Table 1 needs proper reference… For example, for sexual exploitation, please give reference from the parent article?
Different font for lines 325-335
Same for section 4.2 and 4.3 – These sections are discussion rather than results. Please revise.
Discussion is very long – given sections 4.1-4.4 and 5 contain discussion, these should be combined together and re-written.
In abstract, authors note that they included quality appraisal. But this was not described anywhere in the methodology and in the results? Please elaborate what has been done and how it was assessed and what was the result? How did quality appraisal impact the results of the review?
Comments on the Quality of English LanguageEnglish is difficult to understand and follow. Need extensive revision.
Reviewer 4 Report
Comments and Suggestions for Authors
My specific comments and recommendations are included in the attached File.

Suggestions and comments on quality of English language are include in the above attached file
Author Response
Please see the attachement

Round 2
Reviewer 2 Report
Comments and Suggestions for Authors
The first round of edits have strengthened your manuscript and clarified your scientific approach and conclusions. The suggested edits are to finish polishing and clarifying your presentation of the work. I look forward to reviewing the finalized product on this impactful work.
Line 16, add “are”
Line 91, consider changing “gained popularity” to “gained increased attention” to more accurately portray this intention
Line 97, add “to” after “reported”
Line 103, consider changing “will help to provide more shreds of evidence” to “will strengthen collective knowledge base”
Line 110, consider changing “recur to” to “resort to”
Lines 114,117, 119 “abortion” should be plural as it is describing a set of events. Recommending review of full manuscript for whether singular or plural-singular when describing the procedure or within a title/term (such as on lines 108 and 112) and plural when describing procedures occurring
Line 134, add “knowledge about” before “displaced”
Line 254, change “Include” to “are” since all are listed; same at Line 263
Line 276, remove “in this study”, not needed to be clear
Line 428, remove through “that”. Not needed, your factual component is the strong argument.
Line 436, second half of sentence needs to be clarified or made into separate sentence
Line 438, change “may not be” to “are not” to maintain academic objectivity/reduce “argument”
Line 526, remove “From the foregoing”, not needed to be clear
Comments on the Quality of English LanguageThe English language quality has significantly improved. Items in the feedback address items noted that would strengthen further.
